# Salivary Oxytocin Levels in Children With and Without Autism: Group Similarities and Subgroup Variability

**DOI:** 10.3390/jcm14196760

**Published:** 2025-09-24

**Authors:** Eda Yılmazer, Metin Çınaroğlu, Salih Köse, Selami Varol Ülker, Sultan Tarlacı

**Affiliations:** 1Psychology Department, Faculty of Social Sciences, Beykoz University, 34820 İstanbul, Türkiye; edayilmazer@beykoz.edu.tr; 2Psychology Department, Faculty of Administrative and Social Sciences, İstanbul Nişantaşı University, 34398 İstanbul, Türkiye; 3Psychology Department, Institute of Social Science, Üsküdar University, 34662 İstanbul, Türkiye; psk.salihkose@gmail.com; 4Psychology Department, Faculty of Humanities and Social Sciences, Üsküdar University, 34662 İstanbul, Türkiye; selamivarol.ulker@uskudar.edu.tr; 5Neurology Department, NP Hospital, Üsküdar University, 34662 İstanbul, Türkiye; sultan.tarlaci@uskudar.edu.tr

**Keywords:** autism spectrum disorder, oxytocin, saliva, biomarker, children, ELISA

## Abstract

**Background:** Oxytocin (OXT), a neuropeptide involved in social bonding, has been proposed as a potential biomarker for autism spectrum disorder (ASD) due to its role in modulating social behaviors. However, prior studies on peripheral OXT levels in individuals with ASD have yielded inconsistent results, partly due to methodological and developmental variability. This study aimed to compare baseline salivary OXT concentrations between children with ASD and typically developing controls. **Methods:** In this cross-sectional, case–control study, salivary OXT levels were measured in 35 children aged 6–9 years (18 with ASD, 17 controls) using a standardized ELISA protocol. Samples were collected under controlled conditions and analyzed in duplicate. Between-group differences in raw and log-transformed OXT levels were examined using *t*-tests. Subgroup analyses were conducted by sex, and correlations with autism symptom severity (Aberrant Behavior Checklist, ABC) were assessed within the ASD group. **Results:** Children with ASD showed higher mean salivary OXT levels than controls (21.5 pg/mL vs. 14.0 pg/mL), but the difference was not statistically significant (Welch’s t = −1.79, *p* = 0.088). Log transformation of OXT values confirmed the non-significant group difference (t = 1.68, *p* = 0.102). Female participants with ASD had significantly higher OXT than female controls (*p* = 0.048), while no difference was observed among males. OXT levels did not significantly correlate with autism severity (r = −0.04, *p* = 0.88). **Conclusions:** Baseline salivary OXT levels do not significantly differ between children with and without ASD and do not correlate with behavioral symptom severity. However, elevated OXT in females with ASD warrants cautious interpretation and further investigation. Salivary OXT may not be a reliable standalone diagnostic biomarker but could have exploratory value for understanding sex-specific neurobiological profiles in autism.

## 1. Background

OXT is a hypothalamic neuropeptide central to social bonding and affiliative behavior [1]. Given the core social communication difficulties in ASD, disruptions in OXT signaling have been hypothesized to play a role in its underlying neurobiology [2]. This has led to growing interest in examining peripheral OXT levels—particularly in blood and saliva—as potential biomarkers of ASD [3]. Identifying reliable baseline differences between individuals with ASD and neurotypical peers could enhance our understanding of the disorder and support the development of targeted interventions [4]. 

Findings on peripheral OXT levels in individuals with ASD have been mixed. While several studies, particularly in young children, report significantly lower OXT levels in ASD compared to typically developing peers [5,6], others have found no significant differences [7]—or even higher OXT levels in ASD groups [8]. Some evidence suggests these disparities may be more prominent in early childhood and less apparent in adolescence or adulthood [9]. Observed group differences have also been more consistent in male participants [10], though sex-related patterns remain inconclusive [11] due to limited data on females. Overall, the variability across studies underscores that OXT differences in ASD may be subtle and influenced by developmental, methodological, and individual factors.

Several factors likely contribute to the inconsistencies observed across studies on OXT levels in ASD. Methodological variability is a key issue, with differences in biological sample types, assay sensitivity, collection timing, and participant characteristics such as age and clinical profile [12]. OXT’s short half-life and sensitivity to stress or social context mean that its levels can fluctuate significantly [13], making standardized collection procedures critical. Salivary measures, while non-invasive, may yield smaller effect sizes than plasma and raise questions about biological equivalence [14]. Additionally, developmental stage plays a role, as OXT differences appear more detectable in younger children [15]. These challenges have led to calls for more rigorous and standardized approaches, including consistent sampling protocols and focusing on pre-pubertal populations. Clarifying how peripheral measures relate to central OXT activity also remains an important goal. Overall, harmonizing study designs and targeting early developmental windows may help resolve conflicting findings and improve reliability in this area of research.

Saliva has become an appealing medium for OXT research in children, especially those with neurodevelopmental conditions. Unlike blood collection, saliva sampling is non-invasive, stress-free, and well-suited for pediatric populations, including those who may be sensitive to medical procedures [16]. Salivary OXT is considered a biologically meaningful indicator and has been shown to reflect aspects of central OXT function. With advances in assay technology, particularly enzyme-linked immunosorbent assay (ELISA) methods, the reliability of salivary OT measurement has improved, making it viable for large-scale studies [17]. However, relatively few studies have examined baseline salivary OXT in children with ASD [18,19]. While some research has focused on OXT changes following social interaction [20], baseline measures are essential for identifying stable, trait-like differences. Investigating OXT in resting conditions may help clarify whether children with ASD exhibit intrinsically different OXT profiles—an area that remains unresolved due to prior inconsistencies.

In light of these considerations, the present study aimed to systematically compare baseline salivary OXT levels in children with ASD and typically developing peers. We focused on a young pediatric sample (ages 6–9) to capture potential developmental differences in OXT biology. Saliva samples were collected in the morning under standardized conditions and analyzed using a validated ELISA to minimize variability and enhance measurement reliability. In addition to group comparisons, we explored whether individual OXT levels were associated with autism symptom severity, providing insight into potential links between OXT biology and behavioral presentation. Although not a primary aim, we also conducted an exploratory analysis of sex differences; however, these findings are interpreted cautiously due to the limited number of female participants.

## 2. Methods

### 2.1. Study Design

This cross-sectional, case–control cohort study was conducted at NP Private Hospital in İstanbul, Türkiye, a tertiary mental health facility specializing in neurodevelopmental disorders. All participant recruitment, diagnostic evaluations, and saliva sample collections took place on-site at NP Hospital. Saliva samples were collected in the morning hours under standardized conditions and immediately stored at −80 °C. Laboratory analysis of salivary OXT concentrations was performed by Diagen Biotechnology Laboratories (Ankara, Turkey) (www.diagen.com.tr, accessed on 12 July 2025), an ISO-certified molecular diagnostics and ELISA processing center. OXT levels were quantified using a competitive inhibition ELISA kit (Cloud-Clone Corp., Katy, TX, USA, catalog #CEB052Ge). Diagen followed manufacturer protocols under controlled laboratory conditions, including duplicate sample testing and optical density quantification via microplate reader. OXT levels were quantified using a competitive inhibition ELISA kit under controlled laboratory conditions.

### 2.2. Participants

The study included a total of 35 children: 18 children diagnosed with ASD and 17 children without an ASD diagnosis (control group). All children in the ASD group had received a prior clinical diagnosis of autism spectrum disorder (ASD) by a board-certified child psychiatrist or pediatric neurologist in accordance with DSM-5 diagnostic criteria. To confirm diagnostic validity for research inclusion, the Autism Behavior Checklist (ABC) was administered to parents. The ABC is a widely used 57-item screening tool covering sensory, body/object use, relating, language, and social/self-help domains. Scores above 67 are generally considered clinically significant for ASD. In our sample, all children with ASD scored above this threshold (range: 68–97, mean ≈ 75), indicating moderate to severe autism symptomatology. The control group included children without an ASD diagnosis, recruited from the same hospital and community networks. All control participants were screened through clinical evaluation to rule out autism spectrum disorder, and only typically developing children without ASD traits were included. All participants were of similar ages (approximately 7 years old on average in both groups), and no significant age difference existed between the ASD and control groups (mean age 6.94 ± 1.89 vs. 7.00 ± 2.37 years, *t*(33) = 0.077, *p* = 0.94). Gender distribution differed across groups (ASD: 3 girls/15 boys; Control: 7 girls/10 boys), and this imbalance is addressed in the Discussion. This difference in sex ratio was not statistically significant (χ^2^(1) = 2.57, *p* = 0.109). Additional variables such as educational setting and family background were not systematically controlled due to feasibility constraints. However, as part of routine clinical evaluation at NP Hospital, all children in the ASD group underwent standardized cognitive testing using the WISC-R. IQ scores ranged from 77 to 89 (M = 86.2, SD = 3.0), corresponding to borderline intellectual functioning; no child met criteria for moderate or severe intellectual disability. Standardized IQ testing was not conducted for the control group, but clinical evaluation confirmed that none had intellectual disability or learning disorder diagnoses. All participants provided saliva samples for OXT measurement, and informed consent was obtained from parents/guardians under institutional ethical approvals.

Saliva Collection and Processing. Saliva was collected in the morning hours (to minimize diurnal variation) using a passive drool technique. Each child drooled unstimulated saliva into a sterile polypropylene tube, yielding approximately 1–2 mL of sample. Immediately after collection, samples were placed on ice and then stored at −80 °C until assay. Prior to analysis, frozen saliva samples were thawed on ice, gently vortexed, and centrifuged at 1000× *g* for 20 min to remove mucins and any cellular debris. The clear supernatant was used for OXT measurement.

Oxytocin Quantification by ELISA. Salivary OXT concentrations were quantified using a competitive inhibition ELISA kit specific for OXT. The assay procedure was carried out according to the manufacturer’s instructions and can be summarized in the following steps:Plate Preparation. The 96-well microplate was pre-coated with a monoclonal anti-OXT antibody. All reagents and samples were brought to room temperature before use.Standards and Samples. A series of OXT standard solutions (12.35 to 1000 pg/mL) was prepared to generate a standard curve. Saliva supernatant samples were assayed in duplicate. Each well received 50 µL of either standard solution, sample, or zero-standard (blank).Competitive Binding Reaction. Immediately after adding sample or standard, 50 µL of a biotin-labeled OXT Detection Reagent A was added to each well. The plate was gently mixed and incubated for 1 h at 37 °C. During this competitive binding stage, endogenous OXT in the sample competes with the biotinylated OXT for binding to the immobilized antibodies.Washing and Secondary Incubation. Following the incubation, wells were aspirated and washed three times with the provided wash buffer to remove unbound components. Next, 100 µL of Detection Reagent B (HRP-conjugated avidin) was added to each well. The plate was incubated for 30 min at 37 °C, allowing the HRP-avidin to bind to any biotin-OXT that had attached to the plate. Unbound HRP-avidin was then removed by washing five times.Substrate Development. 90 µL of TMB substrate solution was added to each well and incubated in the dark for ~15 min at 37 °C. A blue color developed in proportion to the amount of biotin-OXT (and inversely proportional to the amount of endogenous OXT in the sample). The reaction was stopped by adding 50 µL of stop solution, turning the color to yellow.Measurement. The optical density of each well was immediately read at 450 nm using a microplate reader. The OXT concentration in each saliva sample was then calculated by comparing its absorbance to the standard curve generated from the known standards. Concentrations were expressed in picograms per milliliter (pg/mL).

Assay Sensitivity and Quality Control. The assay’s lower limit of detection was <5.27 pg/mL, and the standard curve covered a dynamic range from ~12.3 pg/mL up to 1000 pg/mL. All samples fell within the linear range of the assay after appropriate dilution if necessary. Intra-assay and inter-assay coefficients of variation were <10% and <12%, respectively, indicating high precision. Each sample was run in duplicate, and averaged values were used in analyses. Any sample with a coefficient of variation > 15% between duplicates was re-assayed to ensure reliability. No samples showed evidence of significant matrix interference as per kit guidelines.

Autism Behavior Checklist. ABC is a standardized screening tool designed to evaluate behaviors associated with autism spectrum disorder (ASD) in children aged 3–12. It consists of 57 items grouped into five subscales: sensory, relating, body and object use, language, and social/self-help skills. In this study, the Turkish-modified version of the ABC was used, which has demonstrated strong reliability and validity in previous research. Items are scored dichotomously (present/absent), with higher total scores indicating greater autism-related behavioral concerns. The ABC was completed by parents of children in the ASD group to assess the severity of behavioral symptoms. This scale has been widely used in educational and clinical settings in Türkiye and supports quantitative assessment of autism traits based on caregiver report [21].

### 2.3. Data Analysis

Group comparisons between the ASD and control children’s OXT levels were conducted using independent-samples *t*-tests. Given that the variance in OXT levels appeared markedly different between groups (with the ASD group showing much higher variability), we first applied Levene’s test for homogeneity of variances. In addition, the distribution of OXT values was examined for normality. Because the raw OXT concentrations were right-skewed and had heterogenous variance, a logarithmic transformation (base 10) of the OXT values was performed for confirmatory analysis to better meet parametric test assumptions. Both the raw and log-transformed data were analyzed; Welch’s *t*-test was used if variances were unequal. We also conducted subgroup analyses by sex (comparing males with ASD vs. male controls, and females with ASD vs. female controls) given the known male predominance in autism and the possibility of sex differences in OXT biology. Finally, within the ASD group, we explored the correlation between salivary OXT concentration and autism symptom severity, as assessed by a standardized rating scale (Autism Behavior Checklist, ABC). Pearson correlation was used for this exploratory analysis. A significance threshold of *p* < 0.05 (two-tailed) was applied for all statistical tests. All analyses have been carried out with Python (version 3.13.4) and SPSS (version 30).

## 3. Results

### 3.1. Oxytocin Levels in ASD vs. Control Groups

The mean salivary OXT concentration in the ASD group was higher than in the control group, but variability was also much greater among children with ASD. Specifically, the ASD group had a mean OXT level of approximately 21.53 pg/mL (standard deviation SD ≈ 16.9), compared to about 14.0 pg/mL (SD ≈ 5.5) in the control group. This difference in means (~7.5 pg/mL higher in ASD on average) did not reach statistical significance at the 5% level when analyzed with a standard *t*-test that does not assume equal variances (*p* ≈ 0.09). In other words, despite a trend toward higher OXT in the ASD group, the large inter-individual spread of values (particularly several high outlier values in the ASD samples) meant that the group difference could not be declared significant with the given sample size. Indeed, Levene’s test confirmed a significant difference in variance between the two groups (*F*1,33 = 10.17, *p* < 0.01), indicating the use of a Welch-adjusted *t* was appropriate.

Because of the notably skewed distribution of OXT levels (a few ASD children exhibited very high concentrations, contributing to the high SD), we re-analyzed the data using log-transformed OXT values. The logarithmic transformation reduced the influence of extreme values and improved homogeneity of variance. After log_10_-transformation, the variance between groups became more comparable, and the overall group difference was still not statistically significant (*t*(33) = 1.68, *p* = 0.102), using a two-sample *t*-test assuming equal variances justified by Levene’s test (*p* = 0.129). Thus, adjusting for the skew and variability did not reveal a hidden significant difference; rather, it confirmed that the mean OXT levels in saliva were broadly similar between the ASD and control children in this sample. We additionally verified the group comparison with a non-parametric test (Mann–Whitney U test) as a sensitivity check, which likewise showed no significant difference in OXT distributions between the two groups (result not shown, *p* > 0.20) (Table 1).

Figure 1 is the Boxplot of raw salivary OXT concentrations (pg/mL) for children with autism spectrum disorder (ASD) vs. typically developing controls. Each box shows the median (line) and interquartile range (box), with whiskers extending to 1.5× IQR; individual points beyond this range are plotted as outliers. In this sample, the ASD group shows a slightly higher median OXT level than controls, but there is substantial overlap between groups. Such overlap is consistent with mixed findings in the literature—for instance, some studies report no significant difference in baseline OXT between children with ASD and controls, even though a meta-analysis of 31 studies suggests overall lower OXT levels in autistic children compared to neurotypical children. In our data, one ASD outlier (upper dot) reflects an especially high OXT value, contributing to greater variance in the ASD group (note the longer upper whisker). This variability mirrors reports that peripheral OXT levels can vary widely in individuals with ASD. Overall, Figure 1 indicates that while the ASD group’s OXT levels tend toward higher values here, the group difference did not reach statistical significance (*p* > 0.05), and there is considerable individual variability.

The group comparison results for both raw and log_10_-transformed salivary OXT concentrations are presented in Table 2. This table summarizes Levene’s test for homogeneity of variance, *t*-test statistics, *p*-values, and effect sizes, providing a concise overview of the differences between the ASD and control groups.

Independent-samples *t*-test results comparing ASD (n = 18) and Control (n = 17) groups on raw OXT values and log_10_-transformed OXT. Levene’s test assesses equality of variances. Effect size is Cohen’s *d*. The raw OXT comparison uses Welch’s *t*-test (unequal variances) due to a significant Levene’s test (*p* < 0.01), whereas the log-transformed comparison assumes equal variances (Levene’s *p* > 0.05).

Figure 2 is the Boxplot of log_10_-transformed salivary OXT concentrations for ASD vs. control children. Here, the log_10_ scale compresses the right-skewed distribution of OXT into a more symmetric range. Skewed distributions are common for hormonal analytes in saliva (including OXT), so a log transformation is often applied to better meet assumptions of normality. In Figure 2, the log-transformed boxplots show more balanced upper and lower whiskers and no extreme outliers, in contrast to Figure 1. The ASD group’s median log-OXT is slightly higher than controls’, reflecting the same trend as the raw data. However, the overlap remains extensive, indicating that even after log transformation the group difference is modest. The use of log_10_ OXT values would be advantageous for parametric statistical analyses (e.g., *t*-tests or ANOVA), since it stabilizes variance and reduces the influence of extreme high values. Indeed, comparing Figure 1 and Figure 2 illustrates how the single high-OXT outlier in the ASD group (Figure 1) is pulled closer to the others on the log scale, no longer appearing as an outlier. This transformation makes it easier to detect any subtle group differences, though in this case the group means remain statistically non-significant (*p* = 0.102), though a numerical trend persists.

Figure 3 is the Overlaid histograms comparing the distribution of raw OXT values vs. log_10_-transformed OXT values across all subjects (ASD and controls combined). The sky-blue bars show the frequency distribution of raw OXT concentrations (in pg/mL), which is notably right-skewed—most children have lower OXT levels on the left, with a long tail toward higher values. In contrast, the orange bars show the distribution of log_10_ (OXT), which appears much closer to a symmetric (approximately normal) shape. The log_10_ transformation effectively normalizes the data by compressing the upper tail. In the raw OXT histogram, a few subjects with high OXT (to the far right) create a long tail, whereas in the log_10_ histogram those same values are pulled into the main body of the distribution. This is evident by the orange histogram’s peak shifting rightward (since log_10_ increases more slowly for high concentrations). The overlaid comparison highlights why a log transform is useful: many statistical tests assume normality, and by transforming OXT levels the data better meet these assumptions. For instance, the raw data’s skew (common in hormone measures) could inflate variance and obscure group differences, whereas the log_10_ data’s more bell-shaped distribution is easier to interpret and analyze.

### 3.2. Sex-Based Subgroup Analysis

As presented in Table 3, given the imbalance in sex ratios and the fact that autism is more prevalent in males, we examined OXT levels within more homogeneous sex subgroups. Interestingly, the small number of female participants showed a noticeable group difference. Among girls, those with ASD (n = 3) had higher salivary OXT than the control girls (n = 7): the mean OXT level for ASD females was 18.0 pg/mL vs. 12.0 pg/mL in females without ASD. This difference was statistically significant (*p* = 0.048 by independent *t*-test) despite the very small sample of girls. By contrast, among male children, the OXT levels did not differ significantly between the ASD (n = 15) and control (n = 10) groups. Male ASD participants had a mean of ~22.23 pg/mL vs. ~15.36 pg/mL for male controls, but this difference failed to reach significance (Welch’s *t*, *p* = 0.20). It is worth noting that variability among the boys with ASD was high (SD ~18.4 pg/mL), which contributed to the lack of a clear difference. No significant interaction effect of group × sex can be formally tested here due to the small female subsample, but the data suggest that the overall group difference observed was driven in part by the female subgroup. Caution is warranted in interpreting this female-specific finding given the low number of girls with ASD in the study. Given the small female subgroup (ASD n = 3; controls n = 7), statistical comparisons were performed on raw OXT values. While the overall group analyses were conducted using log-transformed concentrations to account for variance heterogeneity, log transformation was not applied in subgroup testing because of the very limited sample size, which could yield unstable variance estimates. Thus, sex-specific findings are presented on raw data only and interpreted with caution.

### 3.3. Correlation with Autism Severity

In the ASD group, salivary OXT levels did not show any meaningful correlation with autism symptom severity as measured by the ABC. The Pearson correlation coefficient was *r* = −0.038, indicating essentially no linear relationship (as OXT increased or decreased, there was no consistent change in ABC score), and this correlation was not statistically significant (*p* = 0.881). In practical terms, children with higher or lower OXT concentrations were not systematically more or less severe in their autism symptoms according to the ABC. This lack of association held even after excluding one participant with an extremely high OXT value (analysis not shown), suggesting that within this group of children with ASD, peripheral OXT levels were unrelated to the measured behavioral severity. Additionally, neither age nor sex was found to correlate significantly with OXT levels within the ASD group (all *p* > 0.5 on exploratory analyses), though the sample size may be too limited to detect modest effects presented in Table 4.

Figure 4 is the Scatterplot of salivary OXT concentration (raw, pg/mL) and ABC total scores for the ASD group (n = 18). Each point represents one child with ASD. The black line shows the linear regression fit, with the shaded area representing the 95% confidence interval. There was no significant correlation between OXT levels and ABC scores (Pearson r ≈ −0.04, *p* = 0.88). The regression line is nearly flat, and data points are widely scattered, suggesting that salivary OXT does not predict autism severity as measured by the ABC. This finding aligns with prior studies reporting inconsistent or null associations between peripheral OXT and autism symptom severity.

### 3.4. Cognitive Functioning (IQ)

Within the ASD group, WISC-R IQ scores ranged from 77 to 89 (M = 86.2, SD = 3.0), corresponding to borderline intellectual functioning. No child scored in the range of moderate intellectual disability. IQ was not significantly correlated with salivary OXT concentrations (raw OXT: r = −0.04, *p* = 0.89; log-transformed OXT: r = −0.06, *p* = 0.81). IQ also showed no association with autism symptom severity (ABC scores: r = 0.01, *p* = 0.97). Thus, intellectual functioning did not appear to confound the OXT findings in the ASD group.

## 4. Discussion

In this study, we found that baseline salivary OXT levels did not differ significantly between children with ASD and neurotypical controls (*p* = 0.102), despite a trend toward higher OXT in the ASD group. This result aligns with several reports that failed to detect a robust case–control difference in peripheral OXT concentrations [22,23]. Our finding of no significant group difference contrasts with early studies that reported lower plasma OXT in autistic children [24]. Notably, Modahl et al. [25] observed that typically developing children show an age-related increase in plasma OXT, whereas children with autism failed to exhibit this developmental rise. Such findings gave rise to the “OXT deficiency” hypothesis of autism [26]. A recent meta-analysis of 31 studies indeed concluded that, on average, children with ASD have lower peripheral (blood) OXT levels than neurotypical children [27]. However, exceptions have emerged: a few studies report no difference, and at least one study even found higher salivary OXT in young autistic children compared to controls [28]. In this meta-analysis, few studies show preschool-aged children with ASD and co-occurring intellectual disability had significantly elevated salivary OXT relative to age-matched controls. The authors noted this was an unexpected result given most prior work, and they emphasized that methodological and population differences (assay technique, age range, etc.) likely explain the inconsistency across studies [29]. Thus, our finding of a non-significant trend toward higher OXT in ASD, coupled with substantial overlap between groups, underscores the mixed nature of the existing literature. It suggests that if true baseline differences exist, they are subtle and easily obscured by between-study and inter-individual variability.

A key observation in our data was the markedly greater variability of salivary OXT levels in the ASD group, including several high outliers. Heterogeneity in peripheral OXT is a well-recognized phenomenon [30]. Prior research has shown that the range of OXT levels within each group (ASD or control) is broad, leading to considerable overlap [31]. In one study, some autistic children had as high or higher OXT concentrations as controls, and vice versa [32]. Such variability can dilute between-group differences and may reflect genuine biological heterogeneity as well as technical factors [33]. OXT release is pulsatile and sensitive to context [34], so a single time-point measure may capture transient peaks or troughs unrelated to a person’s typical level [35]. The outliers in our ASD sample could represent children who, for instance, experienced a recent social interaction or stressor influencing OXT release [36], or they might be due to assay noise [37]. Methodological differences in sample collection and assay are critical to consider. Our study used saliva samples; while saliva offers a noninvasive window into OXT physiology [38], it can be methodologically challenging [39]. Salivary OXT levels are typically low and require sensitive assays, with protocols (e.g., whether samples are concentrated via lyophilization or assayed directly) substantially affecting measured values. Minor differences in assay kit characteristics, sample handling, and extraction can produce variability and complicate comparisons between studies [40]. Population factors such as diet, time of day, and emotional state at sampling could further contribute to variability [41]. The developmental stage of participants is another important factor [42]. Our sample consisted of children (mostly in middle childhood, ages ~3–11), an age range across which the OXT system undergoes maturation [43]. It has been demonstrated that OXT levels change with age in a nonlinear fashion during development [44]. In one recent study (in youth with and without OCD), salivary OXT showed associations with age and pubertal stage, with nonlinear increases around puberty, highlighting the need to account for hormonal changes in developmental research [45]. Moreover, as noted above, typically developing children tend to show rising OXT levels with age whereas autistic children may not [46]. Given this dynamic, the inclusion of children at various developmental stages in our cohort could blur group differences if, for example, the ASD group had a different age distribution or a differing maturation trajectory of OXT. Unfortunately, our cross-sectional design and modest sample size do not allow us to dissect age-by-diagnosis interactions. What is clear is that developmental and sampling factors are crucial in interpreting peripheral OXT data. Future studies would benefit from longitudinal designs tracking OXT over developmental milestones, and from standardized sampling protocols (time of day, contextual controls) to reduce extraneous variability.

An intriguing exploratory finding in our study was the sex-specific pattern in OXT levels. When analyzing females and males separately, we found that ASD females had significantly higher salivary OXT than neurotypical females (*p* = 0.048), whereas ASD and control males did not differ. This result should be viewed with caution. First, the female subgroup was very small (only 3 girls in the ASD group versus 7 in the control group), making the analysis underpowered and the *p*-value potentially unstable. Even so, the possibility of a sex-dependent difference in OXT merits consideration in light of broader research. The OXT system is known to interact with sex hormones and to exhibit sexual dimorphism [47]. In the general population, women tend to have higher circulating OXT levels than men [48], and animal studies show OXT receptors are upregulated by estrogen and other gonadal steroids [49]. Thus, one hypothesis is that females (including autistic females) might naturally show higher peripheral OXT [50], which could amplify any group differences among girls. It is also possible that OXT biology in ASD differs by sex [51]: for instance, one gene expression study reported that autistic females had lower OXT receptor (OXTR) expression in certain tissues compared to autistic males and controls, hinting at a complex, sex-specific dysregulation [52]. In our data, the trend of elevated OXT in ASD was essentially driven by the female subset, whereas males with ASD showed a broad range but on average were similar to male controls. We urge caution in over-interpreting this pattern. Given the small numbers and exploratory nature, the female-specific difference could be a Type I error or an artifact of sampling. No prior study, to our knowledge, has conclusively demonstrated a significant sex-by-diagnosis interaction in baseline OXT levels in ASD, so our result would need replication in a larger sample. Nonetheless, it raises interesting questions. If real, higher OXT in females with ASD might reflect a compensatory response or differing regulatory feedback. Females with ASD often present differently from males and may have distinct neuroendocrine profiles; our finding tentatively suggests that the role of OXT in ASD might not be uniform across sexes. Future research should stratify analyses by sex and ensure sufficient inclusion of females, who are underrepresented in ASD studies, to determine if the OXT system’s involvement in ASD indeed varies between boys and girls.

Despite extensive interest in OXT’s role in autism, we found no correlation between salivary OXT levels and autism symptom severity (as measured by ABC scores) in the ASD group. In other words, children with higher OXT were not systematically less (or more) impaired on behavioral indices. This lack of association is consistent with several other studies. For example, Pichugina et al. [53] reported no significant correlation between salivary OXT and symptom severity on the Childhood Autism Rating Scale. The absence of a clear OXT–symptom link in our and other datasets suggests that a single basal OXT measurement may not reflect the complex neurobiology that underlies ASD symptomatology [54]. It is worth noting that some studies have observed correlations, but findings have been mixed in direction. One large study found that higher blood OXT was associated with better social functioning across both neurotypical and ASD children [55]. Intriguingly, that study reported that even within the autism group, those with the lowest OXT had the most severe social deficits, whereas those with the highest OXT had relatively milder deficits. This aligns with the notion of OXT as a “universal regulator” of social ability [56]. On the other hand, a different study observed a paradoxical relationship in autism [57]: plasma OXT was positively correlated with autism severity as measured by the Autism Diagnostic Interview (ADI) social/communication scores (meaning higher OXT in children who had more severe social impairment). The authors interpreted this counterintuitive finding as an indication of dysregulated or compensatory OXT system activity in autism. In our data we saw no significant linear relationship in either direction, which may simply reflect limited power or the noise inherent in peripheral measures. Taken together, the literature suggests that the link between peripheral OXT levels and ASD symptoms is far from straightforward. It is likely moderated by other factors (age, context of sampling, individual biology) and may differ across individuals. These mixed findings also underscore that OXT is not a unidimensional cause or proxy of autism severity [58]. Autism’s etiology is multifactorial, and while OXT pathways may contribute to social functioning, a simplistic “low OXT causes autism” model is inadequate. Our results reinforce that baseline OXT, at least in saliva and in children, is not a reliable indicator of how severe a child’s autism symptoms are.

### 4.1. Clinical Implications

Our findings suggest that baseline salivary OXT levels are not suitable as a standalone diagnostic biomarker for ASD. The lack of a significant group difference and the substantial individual variability indicate that peripheral OXT cannot reliably distinguish autistic children from neurotypical peers. Therefore, clinicians should be cautious about using salivary or plasma OXT as a screening tool or diagnostic aid. However, OXT may still hold clinical value in more targeted applications. Rather than serving as a general marker for ASD, it may help identify subgroups of individuals who could benefit from OXT-based interventions. Prior research has shown that OXT levels are heritable and correlate with social functioning, suggesting that children with naturally low OXT—or relevant genetic variants—might respond more favorably to OXT therapies. This points to a potential role for OXT in personalized treatment approaches, particularly if combined with behavioral or genetic indicators.

The exploratory finding of higher OXT levels in females with ASD also raises the possibility that OXT-related mechanisms—and treatment responses—may differ by sex. If validated in larger samples, such differences could inform sex-specific strategies for intervention or biomarker interpretation.

### 4.2. Limitations

This study has several limitations. The small sample size (18 ASD, 17 controls) reduced statistical power, especially for subgroup analyses such as sex comparisons, where only three girls were included in the ASD group. A few high outliers may also have influenced results. We relied on a single baseline saliva sample, and factors such as time of day, stress, or activity were not systematically controlled. Salivary OXT concentrations are low and assay-dependent, and the extent to which peripheral OXT reflects central activity remains debated. Potential confounders (e.g., pubertal status, diet, co-occurring conditions) were not controlled. The Autism Behavior Checklist (ABC) was administered only to ASD parents, without verification against controls or systematic checks of consistency between parental ratings and professional diagnosis, which may introduce subjectivity. A further limitation is that standardized IQ testing was conducted only in the ASD group, preventing direct between-group comparison. However, all control participants were clinically screened and none showed evidence of intellectual disability. Within the ASD group, IQ scores were largely in the borderline range and did not correlate with OXT levels or autism severity, suggesting that intellectual functioning did not drive the observed results. Finally, participants were not randomly assigned, and variables such as school setting and family background were not matched. Gender distribution was also imbalanced, though sensitivity analyses with a balanced female subsample yielded similar results. These methodological constraints suggest that findings should be interpreted cautiously and replicated in larger, more controlled studies. Moreover, data collection should be expanded to larger, randomly selected samples, or—if purposeful selection is applied—the assumptions and criteria underlying this approach should be explicitly detailed to ensure methodological transparency and reliability.

## 5. Conclusions

This study adds to the growing body of research on peripheral OXT in autism by showing no significant difference in baseline salivary OXT levels between children with ASD and neurotypical peers. Although variability was greater in the ASD group and exploratory findings suggested possible sex-related differences, these results should be interpreted cautiously. Our findings align with prior mixed evidence and highlight the importance of methodological rigor and developmental context in OXT research. While OXT continues to be of interest due to its role in social behavior, the current results suggest it is not a reliable standalone biomarker for ASD. Future studies, particularly those employing longitudinal designs and central biomarkers, are needed to clarify OXT’s relevance to autism’s neurobiology and its potential clinical applications.


## Figures and Tables

**Figure 1 jcm-14-06760-f001:**
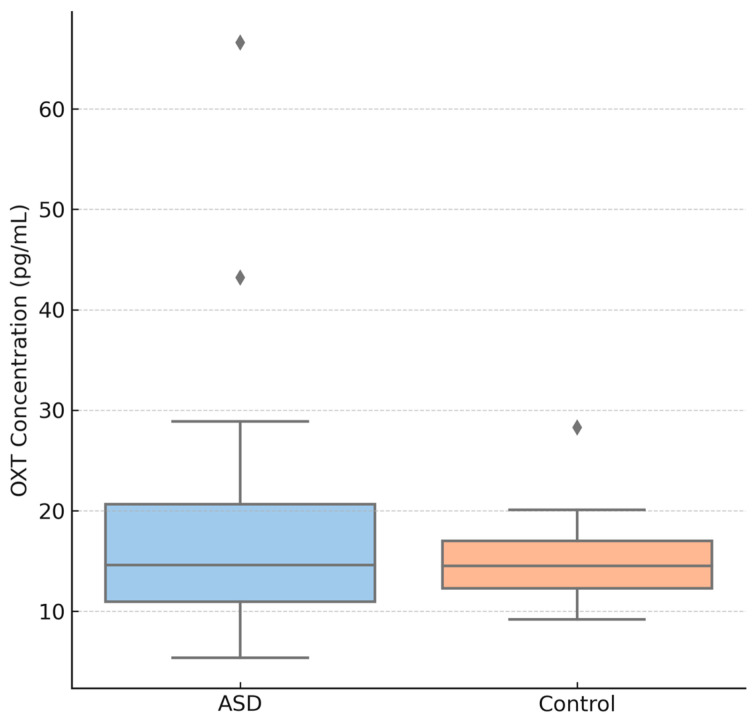
Raw Oxytocin Levels (ASD vs. Control).

**Figure 2 jcm-14-06760-f002:**
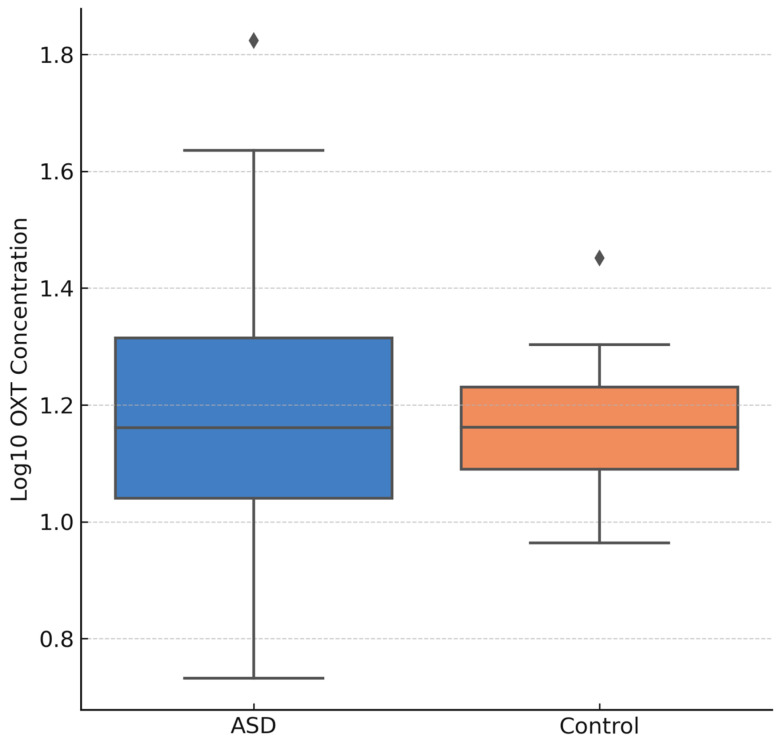
Log_10_-Transformed OXT Levels (ASD vs. Control).

**Figure 3 jcm-14-06760-f003:**
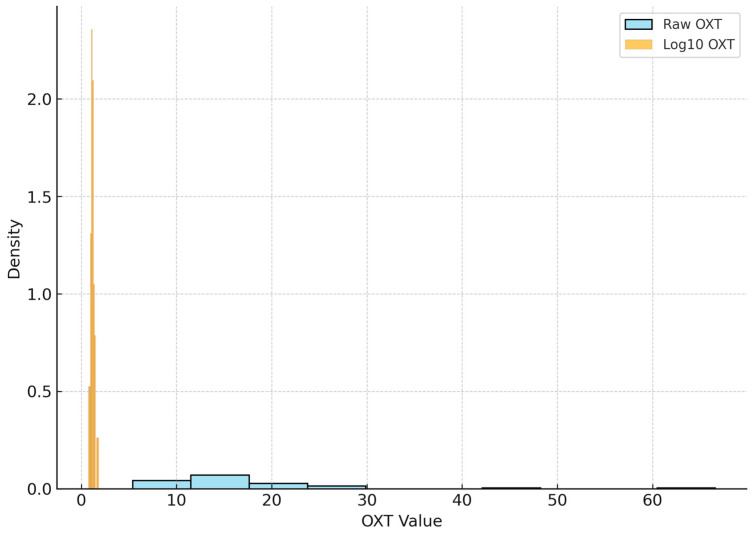
Distribution of Raw vs. Log_10_ OXT Levels.

**Figure 4 jcm-14-06760-f004:**
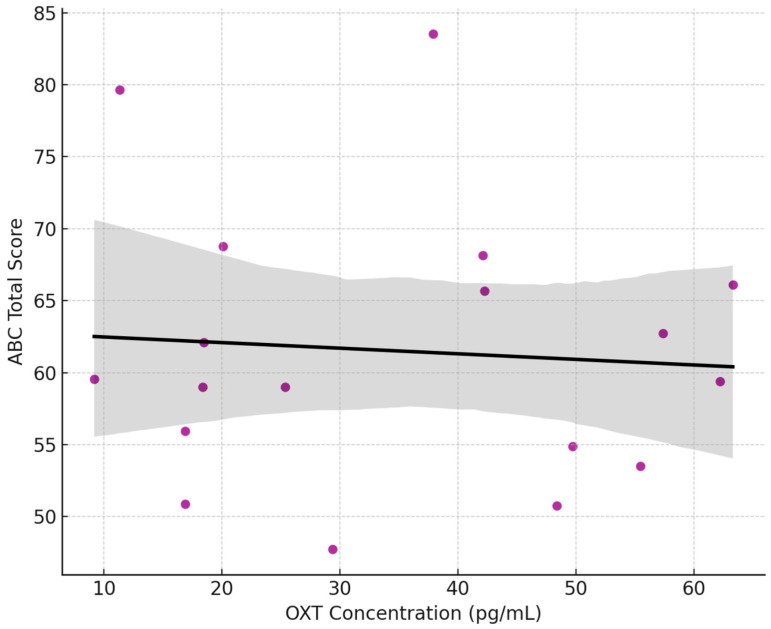
OXT vs. ABC Score in ASD Children.

**Table 1 jcm-14-06760-t001:** Descriptive Statistics for OXT Levels (Raw and Log_10_-Transformed) by Group. Mean, standard deviation (SD), standard error of mean (SEM), minimum, and maximum OXT concentrations are shown in Table 1 for children with autism spectrum disorder (ASD) and neurotypical controls. Raw values are in pg/mL; log-transformed values are unitless (log_10_ of pg/mL).

Group	Data	N	Mean	SD	SEM	Min	Max
ASD	Raw OXT	18	21.53	16.90	3.98	5.4	66.6
	Log_10_ OXT	18	1.21	0.34	0.08	0.73	1.82
Control	Raw OXT	17	14.00	5.55	1.35	9.2	28.3
	Log_10_ OXT	17	1.12	0.14	0.03		

**Table 2 jcm-14-06760-t002:** Group Comparison of OXT Levels (ASD vs. Control).

OXT Measure	Levene’s F (df), *p*	t (df)	*p* (2-Tailed)	Cohen’s *d*
Raw (pg/mL)	F(1,33) = 10.17, *p* = 0.003	*t*(20.8) = −1.79	0.088	0.60
Log_10_ (unitless)	F(1,33) = 2.45, *p* = 0.129	*t*(33) = 1.68	0.102	

**Table 3 jcm-14-06760-t003:** Oxytocin Levels by Sex and Group. Salivary OXT mean ± SD (pg/mL) for ASD and Control children, separated by sex, with independent *t*-test results for group differences within each sex. Female sample size: ASD n = 3, Control n = 7; Male sample size: ASD n = 15, Control n = 10. (Note: Due to small n for females, results should be interpreted cautiously).

Sex	ASD OXT (Mean ± SD)	Control OXT (Mean ± SD)	*t* (df)	*p* (2-Tailed)
Female	18.04 ± 6.41 (n = 3)	12.04 ± 2.19 (n = 7)	−2.33 (df = 8)	0.048 *
Male	22.23 ± 18.38 (n = 15)	15.36 ± 6.82 (n = 10)	−1.32 (df = 19.1)	0.203

* *p* = 0.048 for females indicates a trend toward higher OXT in ASD vs. control girls (equal variances assumed); no significant difference was found in males (*p* > 0.20, Welch’s *t*-test).

**Table 4 jcm-14-06760-t004:** Correlation of Oxytocin and ABC Score in ASD Group. Pearson and Spearman correlation coefficients (r) between salivary OXT concentration (pg/mL, raw values) and Aberrant Behavior Checklist (ABC) total scores in the ASD group (n = 18). Both metrics indicate no significant association.

Correlation	r	*p* (2-Tailed)
Pearson (OXT vs. ABC)	−0.04	0.88
Spearman (OXT vs. ABC)	−0.06	0.81

## Data Availability

The data can be openly accessed on Figshare at the following link: https://figshare.com/s/d76c30ef6edab451aaff (accessed on 11 August 2025). It includes extended descriptive statistics, full group comparison results, sex-specific analyses, correlation data, and detailed methodological procedures.

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
