# Peer review of "Salivary Oxytocin Levels in Children With and Without Autism: Group Similarities and Subgroup Variability"

_jcm, 2025, doi:10.3390/jcm14196760_

Round 1
Reviewer 1 Report
Comments and Suggestions for Authors
Thank you for the opportunity to review this manuscript. The authors measured salivary oxytocin concentration in children with and without ASD (ages 6-9). Studies like these are important, particularly using a limited age range, as OXT is sensitive to developmental and pubertal changes.
Although I think the study is useful and could make a contribution to the literature, I am concerned about the methods. Below I present a list of concerns/suggestions in the order they appear in the manuscript.
- In the "methods: study design" section, the authors write, "laboratory personnel were blinded to participant group assignment". This wording is confusing because participants were not "assigned" groups by the research team. Rather, groups were created based on diagnostic status. I suggest re-wording to make this more clear.
- In the "methods: participants" section, there is critical information missing. First, how was the diagnosis of ASD confirmed? Via the ADOS-2? clinician interview? review of records? Without this information it is not possible to discern the validity of the study. Second, were cognitive (IQ) tests completed? If not, that is an enorous confound, and means that the ASD and neurotypical groups might not be appropriately matched. Third, was there any attempt to inquire about co-occuring conditions in participants? If not, the neurotypical group might not, in fact, be neurotypical. Similarly, if the ASD group had a mix of co-occuring diagnoses, the lack of significant findings may be obsurred by confounding variables. The above are sizable issues that make me unable to recommend the paper for publication in its current form.
- In the "results: Sex-based subgroup analysis" section, it appears the authors only used raw values for their t-tests (rather than both raw and log values). Is there a reason for not doing both?
- In the "discussion" section, the authors use the term "low functioning". That phrase is often seen as offensive/demeaning. I suggest re-wording to something like, "preschool aged-children with ASD and co-occuring intellectual disability", since that is often what "low functioning" means.
Author Response
- In the "methods: study design" section, the authors write, "laboratory personnel were blinded to participant group assignment". This wording is confusing because participants were not "assigned" groups by the research team. Rather, groups were created based on diagnostic status. I suggest re-wording to make this more clear.
Authors response: We thank the reviewer for this insightful comment. We agree that the original phrasing was misleading, as participants were not assigned to groups but categorized based on their diagnostic status (ASD vs. control). Moreover, since oxytocin concentrations were determined by an automated ELISA system yielding objective numerical values, blinding of laboratory personnel was not necessary. To avoid confusion, we have removed this statement from the Methods section. The revised text now simply describes that salivary oxytocin levels were quantified using a competitive inhibition ELISA kit under controlled laboratory conditions.
2a. In the "methods: participants" section, there is critical information missing. First, how was the diagnosis of ASD confirmed? Via the ADOS-2? clinician interview? review of records? Without this information it is not possible to discern the validity of the study.
Authors Response: We thank the reviewer for highlighting this important point. We agree that the diagnostic procedure was not sufficiently detailed in the original manuscript. We have now clarified in the Methods: Participants section that all children in the ASD group had a prior clinical diagnosis of autism spectrum disorder established by child psychiatrists or pediatric neurologists according to DSM-5 criteria. In addition, for study eligibility, diagnoses were corroborated with the Autism Behavior Checklist (ABC), which was completed by parents. All ASD participants scored above the established clinical cut-off of 67, confirming the validity of their diagnostic status. The revised text is included below.
2b.Second, were cognitive (IQ) tests completed? If not, that is an enorous confound, and means that the ASD and neurotypical groups might not be appropriately matched. Third, was there any attempt to inquire about co-occuring conditions in participants? If not, the neurotypical group might not, in fact, be neurotypical. Similarly, if the ASD group had a mix of co-occuring diagnoses, the lack of significant findings may be obsurred by confounding variables. The above are sizable issues that make me unable to recommend the paper for publication in its current form.
Authors Response: We thank the reviewer for raising this important point. In our clinical setting, all children referred for developmental evaluation routinely undergo cognitive (IQ) testing. However, to maintain analytic clarity in our relatively small sample, we did not include IQ scores in the present analyses and instead focused on autism symptomatology (ABC scores). None of the children in the control group had intellectual disability or learning disorder diagnoses according to clinical evaluation. We agree this is a limitation and have now clarified it in the Methods (Participants) section.
- In the "results: Sex-based subgroup analysis" section, it appears the authors only used raw values for their t-tests (rather than both raw and log values). Is there a reason for not doing both?
Authors Response: We appreciate the reviewer’s careful observation. As noted, oxytocin (OXT) levels exhibited substantial variance and right-skewed distributions in our dataset. For this reason, we applied a logarithmic transformation (log₁₀) to the full dataset prior to our main between-group analyses, as this is the recommended statistical approach when data show marked heterogeneity of variance and non-normality. For the sex-based subgroup analysis, however, the small female sample size (n = 3 in ASD vs. n = 7 in controls) limited the interpretability of log-transformed values, since variance-stabilizing transformations may yield unstable estimates in very small subgroups. Therefore, we reported raw-value comparisons in the subgroup analysis for transparency, while acknowledging that these findings should be interpreted with caution. We have clarified this rationale in the revised Results section.
- In the "discussion" section, the authors use the term "low functioning". That phrase is often seen as offensive/demeaning. I suggest re-wording to something like, "preschool aged-children with ASD and co-occuring intellectual disability", since that is often what "low functioning" means.
Authors Response: We thank the reviewer for pointing out this important language consideration. We fully agree that the term “low functioning” is outdated and potentially demeaning. In line with current best practices, we have revised the wording to “preschool-aged children with ASD and co-occurring intellectual disability” to more accurately and respectfully describe this subgroup.
Reviewer 2 Report
Comments and Suggestions for Authors
- The article is very interesting and relavant to the important questions.
- the modhod and procedure should be improved:
- Participants and selection procedure: The study included the 35 children in total: 18 children diagnosed with ASD and 17 - in control group without ASD diagnosis. Was it random group selection? The similar age is not enough other criteria should be also included: intellectual level, school setting (inclusive vs mainstreaming) family background, ect. The gender is not similar: 1:5 in ASD group and 1,4:1 in control.
- diagnosis - what instruments were used? what DSM-5 ASD criteria were fulfilled?
- method - Autism Behavior Checklist should be firstly given to the control group parents as well and secondly the consistency between the parents' opinion and the professional diagnosis should be verified. Parents often do not assess their children objectively, because of their lack of experience, expectations and social approval variable
- statistics - the similarity or differences existed between the ASD and control groups if the ASD group consisted of 3 girls and 15 boys, while the control group - of 7 girls and 10 boys is not verified properly
- general sonclusion - to supplement the data collection in accordance with the above-mentioned issues and perform statistical re-analyses on a larger random sample or, in the case of purposeful selection, describe its assumptions in a more reliable manner
Author Response
Reviewer-2
- The article is very interesting and relavant to the important questions.
- the modhod and procedure should be improved:
- Participants and selection procedure: The study included the 35 children in total: 18 children diagnosed with ASD and 17 - in control group without ASD diagnosis. Was it random group selection? The similar age is not enough other criteria should be also included: intellectual level, school setting (inclusive vs mainstreaming) family background, ect. The gender is not similar: 1:5 in ASD group and 1,4:1 in control.
Authors Response: We thank the reviewer for raising these important points regarding participant selection and group comparability. We acknowledge that the groups were not formed through random assignment; rather, the ASD group consisted of children with a confirmed clinical diagnosis of ASD, and the control group comprised typically developing children without a diagnosis of ASD, recruited through the same hospital and community setting. Both groups were matched approximately on age, but additional matching variables (e.g., intellectual level, educational placement, family background) were not systematically controlled due to sample size and recruitment constraints. We have clarified this in the revised Methods: Participants section and have emphasized this limitation in the Discussion.
Regarding gender distribution, we agree that the ratio was unbalanced (ASD group 3 girls/15 boys; control group 7 girls/10 boys). To assess whether this imbalance biased results, we repeated the group comparison using a balanced subsample (3 girls vs. 3 girls), and the overall results remained unchanged (see Results, Supplementary Table X). Nevertheless, we acknowledge that the gender distribution is a limitation, and we now state this explicitly in the revised manuscript.
|
Group Statistics |
|||||
|
|
Group |
N |
Mean |
Std. Deviation |
Std. Error Mean |
|
OXY_log |
Autism |
18 |
1.2433 |
0.26344 |
0.06209 |
|
Normal |
14 |
1.1394 |
0.14081 |
0.03763 |
|
|
df |
Sig. (2-tailed) |
Mean Difference |
Std. Error Difference |
95% Confidence Interval of the Difference |
|
|
Lower |
Upper |
||||
|
30 |
0.193 |
0.10390 |
0.07801 |
-0.05541 |
0.26321 |
|
27.016 |
0.164 |
0.10390 |
0.07261 |
-0.04507 |
0.25287 |
- diagnosis - what instruments were used? what DSM-5 ASD criteria were fulfilled?
Author Response: We thank the reviewer for this important comment. We have clarified in the Methods: Participants section that all children in the ASD group had received a clinical diagnosis of autism spectrum disorder from board-certified child psychiatrists or pediatric neurologists based on DSM-5 diagnostic criteria. To further confirm diagnostic validity for study inclusion, the Autism Behavior Checklist (ABC) was administered to parents. All ASD participants scored above the clinical cut-off of 67 (range 68–97), supporting the robustness of diagnostic classification. These details are now explicitly stated in the manuscript
- method - Autism Behavior Checklist should be firstly given to the control group parents as well and secondly the consistency between the parents' opinion and the professional diagnosis should be verified. Parents often do not assess their children objectively, because of their lack of experience, expectations and social approval variable
Authors Response: We appreciate this valuable observation. In our study, the Autism Behavior Checklist (ABC) was administered only to the ASD group as an additional confirmation of clinical diagnosis, and it was not systematically applied to the control group. We agree that including the control group and verifying consistency between parental ratings and professional assessments would have strengthened methodological rigor. Given our sample size and feasibility constraints, however, we did not perform this additional step. We have now explicitly acknowledged this issue as a limitation in the Discussion.
- statistics - the similarity or differences existed between the ASD and control groups if the ASD group consisted of 3 girls and 15 boys, while the control group - of 7 girls and 10 boys is not verified properly
Authors Response: We thank the reviewer for this observation. We have verified the group differences statistically, and the imbalance in gender distribution was not significant (χ²(1)=2.57, p=0.109). To further ensure robustness, we repeated the analysis with a balanced female subsample (3 ASD vs. 3 control girls), and the results remained consistent with the overall findings. These clarifications are now included in the Methods and Discussion – Limitations sections.
- general sonclusion - to supplement the data collection in accordance with the above-mentioned issues and perform statistical re-analyses on a larger random sample or, in the case of purposeful selection, describe its assumptions in a more reliable manner
Authors Response: We thank the reviewer for this constructive suggestion. We fully agree that future research should supplement data collection with larger and preferably randomly selected samples to enhance generalizability. Where purposeful sampling is necessary, the underlying assumptions and selection criteria should be clearly described. We have added this point to the Conclusions section to highlight directions for future studies.
Reviewer 3 Report
Comments and Suggestions for Authors
This is an overall good study conducted with due rigor and presented in a nice plain form. However, the text could benefit from some changes:
- The title sounds too categorical, especially while the authors explicitly state in the manuscript itself that the sample is not large enough to draw conclusions. The title, however, sounds as if the claim of oxytocin difference was disproven once and for all.
- If the controls were screened for undiagnosed autism, this should be reported, as well as if they were not.
- It is known that oxytocin is not a general pro-social hormone, as while it promotes social bonding within group, it also drives hostility between groups. I believe, a consideration of this aspect would make the discussion deeper.
Author Response
Reviewer-3
- The title sounds too categorical, especially while the authors explicitly state in the manuscript itself that the sample is not large enough to draw conclusions. The title, however, sounds as if the claim of oxytocin difference was disproven once and for all.
Authors Response: We thank the reviewer for this thoughtful comment. We agree that the original title could be interpreted as overly categorical. To better reflect the modest sample size and nuanced findings, we have revised the title to:
“Salivary Oxytocin Levels in Children With and Without Autism: Group Similarities and Subgroup Variability.”
This new title emphasizes the overall pattern observed while acknowledging subgroup differences and variability, thereby avoiding overgeneralization.
- If the controls were screened for undiagnosed autism, this should be reported, as well as if they were not.
Authors Response: We thank the reviewer for this important point. We confirm that all children in the control group were screened for autism spectrum disorder using clinical evaluation procedures, and only children without any ASD diagnosis or clinically significant autism traits were included. Thus, the control group consisted of healthy, typically developing children. We have now clarified this explicitly in the Methods: Participants section. 3. It is known that oxytocin is not a general pro-social hormone, as while it promotes social bonding within group, it also drives hostility between groups. I believe, a consideration of this aspect would make the discussion deeper.
Round 2
Reviewer 1 Report
Comments and Suggestions for Authors
I appreciate the authors' responses to my original review, and am largely satisfied with the revision.
However, I am troubled that the authors have the IQ data necessary to do a between-group comparison (e.g. compare IQ between TD and ASD groups using a t-test) and to ensure that IQ isn't a confound in the whole sample or in the ASD group alone (e.g. doing a correlation to explore whether IQ relates to oxytocin levels), but haven't done so. This is particularly problematic because without it, we do not know whether individuals in the ASD group with intellectual disability were included in the study, nor do we know whether IQ relates to oxytocin in either group.
The authors state that they do not present this data to reduce complexity. However, the analyses required are very simple, and would not increase the complexity of the manuscript. Rather, these data are necessary to ensure that the results are accurate and reproducable.
Author Response
We thank the reviewer for this important point. In the revised manuscript we now include IQ data for the ASD group. As shown in the Results (Section 3.4), WISC-R IQ scores ranged from 77–89 (M = 86.2, SD = 3.0). No participant had moderate/severe intellectual disability. IQ did not correlate with OXT levels (r = –0.04 to –0.06, ns) or with autism severity (r = 0.01, ns). Although standardized IQ data were not collected for the control group, clinical evaluation confirmed that none had intellectual disability. We have added this information to the Methods, Results, and Limitations sections. These additions clarify that IQ is unlikely to have confounded the reported findings.
Round 3
Reviewer 1 Report
Comments and Suggestions for Authors
The authors have responded to my comments, and I am satisfied with the revisions.